# The Use of Wearable Technology in Providing Assistive Solutions for Mental Well-Being

**DOI:** 10.3390/s23177378

**Published:** 2023-08-24

**Authors:** Reham Alhejaili, Akram Alomainy

**Affiliations:** 1School of Electronic Engineering and Computer Science, Queen Mary University of London, London E1 4NS, UK; 2Department of Computer Science and Artificial Intelligence, College of Computer Science and Engineering, University of Jeddah, Jeddah 23218, Saudi Arabia; 3Antennas and Electromagnetics Research Group, School of Electronic Engineering and Computer Science, Queen Mary University of London, London E1 4NS, UK; a.alomainy@qmul.ac.uk

**Keywords:** wearable devices, mental health, stress, anxiety, EEG, HRV, EDA

## Abstract

The main goal of this manuscript is to provide an extensive literature review and analysis of certain biomarkers, which are frequently used to identify stress, anxiety, and other emotions, leading to potential solutions for the monitoring of mental wellness using wearable technologies. It is possible to see the impacts of several biomarkers in detecting stress levels and their effectiveness with an investigation into the literature on this subject. Biofeedback training has demonstrated some psychological effects, such as a reduction in anxiety and self-control enhancement. This survey demonstrates backed up by evidence that wearable devices are assistive in providing health and mental wellness solutions. Because physical activity tracing would reduce the stress stressors, which affect the subject’s body, therefore, it would also affect the mental activity and would lead to a reduction in cognitive mental load.

## 1. Introduction 

Due to the substantial increase in the world population and the financial and resource strains it is adding to healthcare provision, physicians and researchers need to suggest new techniques and tools that could simplify their roles and practices and contribute to providing patient/user with effective assistance [1]. This would help reduce the level of strain, tension, and anxiety, which have increased significantly due to elevated levels of performance and anxiety. Researchers have suggested updated approaches to assist healthcare professionals and doctors in identifying various vital factors and criteria and taking different human body measurements to assess the degree of stress and distress a person might be experiencing [2]. The critical purpose of the research analysis is to undertake a comprehensive systematic literature review. In order to achieve this, numerous electronic sources and academic journals were reviewed in depth, and several keywords and research terms were used to classify relevant publications, journals, and research papers related to wearable devices [3]. 

The obstacles and challenges, which healthcare workers encounter, can be substantially minimized with the implementation of emerging technology and recent developments [4,5]. The authors believe this study identifies the research gaps and challenges and provides a clearer picture of the current state of the art and potential research directions. More than 50 studies from 2016 to 2021 are examined in our analysis. For example, wearable solutions have been incorporated into numerous studies with potentially positive results to provide a detailed picture of Parkinson’s disease patients’ well-being [4]. On the other hand, this manuscript is more focused on the subject of well-being and the innovative and emerging wearable technologies, which are expected to be used to diagnose stress and anxiety in workplace environments beyond specific case studies or scenarios, providing a more comprehensive picture. 

Numerous studies have been conducted with encouraging outcomes to examine the effectiveness of devices, including wearable gadgets, and have shown improvements in diseases and effects. Wearable technology has made remote monitoring and treatment of illnesses feasible [6]. The research study focuses on wearable, non-invasive technologies, which measure human stress levels utilizing numerous parameters [6]. A study by Martinez et al. reported a dramatic increase in patients with cognitive load and anxiety in recent years [7]. The study’s author also addressed stress as one of the states, which have become epidemiologically more prevalent in most nations worldwide [8]. Wearable sensor-based technologies can sense different parameters of a person, which can be viewed as signs of stress within the individual. These indications, often used for stress and stress detection, are the heart rate, pulse rate, pupil dilation, skin temperature, and electrodermal activity in the human body [9]. The research further illustrates that, although low stress levels have always benefited the human body, the risk factors, challenges, and difficulties should be considered when making choices or performing everyday activities. If a person is continuously under challenging times, this could have a detrimental effect on human physiology, cognition, and neurocognitive behavior. Constant stress can lead to long-term illnesses, such as anxiety, depression, and rapid aging [10]. Healthcare providers should design a management plan for their patients with the help of such type of detection.

The appearance of stress within an individual contributes to additional adverse effects and repercussions on the individual, including anxiety, depression, and panic attacks. However, with the aid of specific technologies, which monitor and recognize the physiological and mental changes within a person naturally, it is possible to improve a person’s quality of life through the early identification of problems. These devices also help determine the underlying stimuli, which generally cause disruptive behavior and cognitive problems. Healthcare providers should design a management plan for their patients with the help of such type of detection.

The rest of the paper will provide an in-depth analysis of the topic, starting with the motivation and background which led to the research. The literature review will explore various sources, including academic journals and research papers, to gain a better understanding of wearable devices and stress detection technologies. The study will also provide a survey analysis to investigate the level of stress among healthcare professionals and doctors. The discussion will highlight the research’s findings and implications for the healthcare industry, while the conclusion will summarize the paper’s main points and provide recommendations for future research.

### 1.1. Motivation and Background 

Body odor helps detect fear and anxiety in individuals, and specific biomarkers, such as the heart rate, skin conductivity, and oxygen saturation, can be used to assess stress and anxiety in patients. Hui and Sherratt [11] evaluated these vital signs and biomarkers to analyze the changing mental states and conditions. In contrast, Happy and Routray [12] used image processing to detect emotional changes by assessing facial expressions, and Li et al. [13] utilized a similar method but added an analysis and assessment of electroencephalography reports. Using these reports and facial expressions, they were able to determine whether there was a connection between changes in facial expressions and changes in brain activity and electrical nerve impulses. EEG assessment is considered one of the most beneficial methods for detecting mental states and conditions, and research studies utilizing this method are the focus of this literature review [14]. 

### 1.2. Research Scope

Due to mental health disorders and complications among patients, there is a greater need for doctors, psychiatrists, psychologists, and healthcare professionals to assist and treat mental issues [15]. Therefore, researchers need to consider existing theories, inventions, and concepts related to wearable devices, which can detect cognitive mental parameters in these conditions. This systematic literature review is designed to provide readers and future researchers with knowledge about wearable devices used to monitor and assist with mental well-being, which can be helpful in the future [15]. 

### 1.3. Aims of the Research

The primary focus of this research study is to investigate how wearable devices can capture biomarkers, which aid in detecting and potentially assisting with mental well-being. The study aims to conduct a literature review on this topic to observe the effects of different biomarkers in detecting stress levels and their effectiveness. Effective remedies and solutions have been developed based on the type of device used to monitor mental conditions and emotional parameters.

### 1.4. Significance of the Research

Given the rapid increase in mental and psychological disorders and complications faced by individuals and the lack of resources to provide medical assistance to such patients, it is crucial to conduct multiple research studies analyzing the beneficial effects of wearable devices. Healthcare practitioners and individuals can use these devices to assess their mental state and condition during their daily routines. The study’s contribution lies in the advancements of sensor-based wearable technologies, which can provide an effective solution for mental well-being.

## 2. Literature Review

### 2.1. Overview

According to a research study conducted by the Organization for Economic Cooperation and Development, i.e., OECD, it has been shown that due to the world’s rapidly growing population, there is an increased need and demand for further resources in terms of healthcare providers and practitioners [16]. However, due to the shortage of such professionals, researchers need new innovative technologies and solutions, which can easily measure the patients’ physiological and mental health conditions [17]. Such types of emerging technologies are termed as quality-of-life technologies (QOTS). Another research study supported this fact and explained that the misclassification of an acute disease condition as chronic by electrochemical sweat biomarker sensors could cause significant psychological, emotional, and financial stress among patients [18]. In addition to monitoring the vital and psychological parameters of the patients, some of these innovative technological devices are being researched, which will help elevate and modify the patients’ moods and emotional conditions [18]. The following section will outline an explanation of the wearables, sensors, and signal detection biomarkers associated with stress factors and an assessment of emotions and mental well-being.

### 2.2. Machine-Learning Methodologies for Identifying Stress Factors in Physiological Data Captured through Wearable Devices

While current researchers focus on real-life environments, earlier works on stress detection were performed in laboratory environments. The most widely used physiological signals for detecting stress levels include the accelerometer, heart activity (HR), and electrodermal activity (EDA). In laboratory environments, HR and EDA combined are known to have the best performances. Cho et al. [19] showed that EEG signals achieved 89% accuracy in a four-class stress classification. However, the currently used EEG measuring devices are obtrusive and irrelevant to daily routines. Wearable biosensors are a basic mediator for physiological signal records, and they can be utilized for daily real-life situations, clinical applications, and fundamental exploration studies, as they allow users to continue their regular activities without interruption [20,21]. Wearable biosensors offer excellent information and data, which are adequately portrayed, detailed, timely, relevant, complete, and accurate, and the extracted data retain appropriate related evidence for supporting decision-making processes [22].

Nevertheless, using wearable biosensors in real-life experimentation presents numerous challenges regarding the reliability and usefulness of the measurements for extracting emotions. A sufficient sampling frequency is required to depict signals correctly, and the sensor must be placed properly to record physiological signals while avoiding ambiguities accurately. Raw signals normally have many fluctuations due to the oscillations of the human body’s physiological status [23,24]. Even if the sensor is placed properly and the fluctuations are inevitably recorded, it is necessary to filter the raw sensor signal to remove noise. This would allow for increased efficiency in detecting stress. Several filters, such as wavelet decomposition, Wiener filter, median filter, Butterworth filter, and Kalman filter, can be used for filtering noise. Selecting an ideal filter depends on the type of noise, the features to be extracted, and the nature of the signal [23].

Lastly, it is worth mentioning that it is inappropriate for investigators to control environmental factors in real-life cases. Therefore, isolating the impact of a stimulus is challenging. However, despite these challenges, several works have tried to detect stress levels using wearable devices, as presented in this paper.

Stress-related illnesses have been identified as the second most common type of illness after musculoskeletal illnesses, which may also be caused by stress-related issues [22]. The type of illness a person experiences can sometimes be correlated with the level and type of stress they are facing, including emotional distress, headaches, low hormone levels, digestive problems, and over-arousal, among others [23]. Over-arousal can cause severe harm to those with underlying conditions, potentially leading to heart attacks, panic attacks, and even sudden death in some cases. Chronic stress is commonly diagnosed in patients with hypertension, coronary heart disease, irritable bowel syndrome, gastroesophageal reflux disease, generalized anxiety disorder [22], and depression [24]. Stress can also lead to more downtime for individuals, causing them to experience lower energy levels, which can ultimately affect the economy. The global economy suffers when people experience downtime related to work-related and personal stress [25]. Various studies across the European economy and its workforce indicate that up to half of all working days may be lost due to stress-related downtime [26,27].

### 2.3. Exploration of Assistive Solutions for Mental Well-Being

An ideal scenario would involve the absence of stress or the removal of stress-inducing factors and conditions, but this is rarely achievable. Long-term stress can result in severe consequences and must be treated with care. Since the treatment does not have a fixed endpoint or path to follow, it necessitates constant monitoring and adjustment over a prolonged period, which may span several months or years. It is best to address stress early to reduce its long-term physical and mental consequences [20], such as attitude changes, mood swings, or loss of taste. Although several tools exist to monitor stress-related vital signs, no direct measurement system exists, and we can only treat the symptoms. The medical community may gather and analyze data using smartwatches and portable health monitors, which are novel tools.

The emergence of computer programming, data science, and powerful computers has made it feasible to develop complex algorithms, which can provide extensive real-time data used to evaluate a patient’s condition. Data science has revolutionized the use of data to address today’s challenges and may be applied in a variety of fields [27]. During work or stressful interactions, data may be collected and examined to assess their impact on individuals [22].

### 2.4. Assessment of Emotions

Emotions are considered one of the most essential, yet important, aspects of everyone’s life, which need to be studied by researchers, as they are primarily responsible for generating multiple diseases, disorders, and complications among individuals. Wearable technology can detect people’s stress levels as they go about their everyday activities and determine the most frequent triggers, which put people in stressful situations. However, this approach can also be responsible for causing various stressful conditions and situations for a person [23]. Several studies have shown the effectiveness of wearable technologies and devices in monitoring a person’s stress levels. These studies have provided a descriptive report of all the innovative technologies used to monitor stress levels. However, most of these studies only focused on specific articles, which provided detailed information about the research questions. They reported that the focus and emphasis of devices relied primarily upon EEG and CPS systems, which can monitor a person’s stress levels [24].

All these research studies were concluded and compiled in order to analyze some of the significant patterns of evolution of wearable sensor-based devices, which help monitor a person’s stress levels. Numerous research articles and experiments discovered among all the studies included in this systematic literature review showed that monitoring wearable devices for stress measurement employed in a controlled environment are also considered inclusion criteria [25]. Special conditions and situations were explained to the participants; under controlled circumstances, the devices monitored and captured stress levels. In such conditions, observing the stress levels in difficult situations becomes challenging. Although multiple stimuli were provided to the participants to monitor their stress levels even in challenging situations, such studies are not considered beneficial, as the methods failed to suggest the results, which might be obtained under normal circumstances and in the everyday lives of individuals [26].

The core objective of modifying wearable devices is to enable them to analyze the individuals’ stress levels when exposed to their daily routines. This way, we can analyze some of the most common stimuli and why people often find themselves in stressful situations. Moreover, this criterion also helps us analyze and investigate only those research studies where real-life examples and testing were performed. This criterion is set to include literature reviews and research from different authors, so that the effectiveness of the devices can be observed and analyzed in real-life situations. In most studies, the research is conducted within workplaces, at home, and for elderly patients in nursing homes [27,28,29].

### 2.5. Types of Physiological Signals and Markers

Measuring specific vital signals, such as the heart rate, galvanic skin response (GSR), and body temperature, is essential to gauging people’s stress levels. These parameters are detected through various methods, including electrocardiography (ECG), which monitors heart activity and helps analyze the changes in heart rate under different conditions and emotional stimuli. Research study suggests that monitoring these parameters can identify significant stimuli responsible for causing stress [3,30].

It is recommended to perform photoplethysmography (PPG) to detect changes in blood volume in a chosen artery, which can help identify the changes in blood volume. Additionally, monitoring an individual’s galvanic skin response (GSR) is crucial, as it enables the quantification of variations in skin conductance.

Constant monitoring of stress levels requires performing an electroencephalogram (EEG) on the patient, which records and tracks the brain’s electrical activity [31]. Another essential factor is monitoring the patient’s respiration rate (RSP), which provides insights into their stress levels. When conducting examinations and tests, it is crucial to identify the patient’s pre-existing physiological and psychological conditions. This differentiation can determine whether the responses obtained from the tests and examination reports are due to the stress stimuli experienced by the individual or their pre-existing medical conditions [11].

### 2.6. Devices Used for the Detection of Stress Levels

Saganowski et al. conducted a systematic literature review by analyzing and compiling the results and experiments from multiple researchers. The study aimed to detect vital parameters within individuals, including GSR, heart rate, pulse rate, heartbeat intervals, respiratory rate, and body temperature. The study established discrete values for each dimension, such as joy, sadness, stress, calm, happiness, boredom, or neutrality, which were used to identify a person’s current mental status and condition. However, the study noted that other dimensions might be observed during analysis and investigation only in research studies where real-life testing and examples have been performed [32].

Various devices, such as Empatica E4, Microsoft Band 2, Samsung Gear S, Body Media Sense Wear Armband, Neurosky Mind Wave, and XYZ Life Bio Clothing, have been selected for monitoring individual vital signs and parameters. These wearables monitor physiological functions, such as the heart rate, breathing rate, and respiratory care, enabling easy analysis of the brain’s electrical activities. The study highlights the availability of commercially available and comfortable wearables on the market, which can collect physiological signs and signals to monitor participants’ stress levels and anxiety conditions [13]. 

Furthermore, some articles have proposed self-made devices, but their efficacy and effectiveness require further investigation. Each device is responsible for measuring specific sensors and parameters in an individual. The past years have witnessed explosive growth in wearable devices, which aim to monitor a broad spectrum of human physiology and behavior, employing diverse sensors and technologies to gather data that shed light on users’ health, well-being, and performance. The Empatica E4 is a versatile wearable tracking PPG, GSR, and BT, dissecting body temperature and skin responses to activities and stimuli. Similarly, the Microsoft Band 2 employs PPG, GSR, and BT sensors, contributing to stress, activity, and emotional monitoring. The Samsung Gear S specializes in heart rate observation via a heart rate sensor, optimizing exercise and cardiovascular health assessment. Meanwhile, the BodyMedia SenseWear Armband employs multiple sensors for comprehensive analysis of vital signs, such as body temperature, skin response, and activity levels. The NeuroSky MindWave employs EEG technology to discern brainwave activity, enabling monitoring of mental states and focus. The XYZ Life Bio Clothing virtually tracks EEG signals to unveil cognitive states and stress levels discretely. These devices exhibit differences in the detected signals and primary focuses, catering to physiological metrics. While some, such as Empatica E4 and Microsoft Band 2, cover a broad spectrum, others, such as Samsung Gear S and BodyMedia SenseWear Armband, concentrate on specific aspects, such as the heart rate and activity. The NeuroSky MindWave and XYZ Life Bio Clothing excel in EEG monitoring, offering insights into brainwave patterns and mental states. Wearables have transformed data acquisition from our bodies and minds, addressing diverse needs, from fitness to emotional well-being. As technology advances, these devices are poised to deliver even deeper insights [33].

### 2.7. Self-Assessment of Emotions

Emotions play a dual role, which is beneficial when they enhance decision making and motivate appropriate behaviors but potentially detrimental when emotions are of inappropriate intensity, frequency, or duration, necessitating emotion regulation to prevent harm to oneself or others. Emotion recognition systems help in this regard. A positive goal, such as reducing sadness or promoting a healthy lifestyle, is essential to regulate emotions effectively. Emotional regulation can be intrinsic, with individuals managing their own emotions. Additionally, emotions have been understood through two main approaches—discrete or categorical emotion states and continuous or dimensional emotion space—with researchers categorizing primary emotions, such as fear, grief, love, and rage, based on intense physiological changes. However, self-report questionnaires’ reliability and emotion awareness pose challenges in training supervised machine-learning algorithms for emotion recognition. While laboratory experiments establish the ground truth, ecological momentary assessment using self-reports is conducted outside the laboratory, but subjectivity influenced by individual, cultural, and gender factors may affect accuracy. Obtaining frequent real-time self-reports during daily routines proves challenging, leading to delays in labeling and potential information loss. The demand for labeled data for robust models increases the reliance on self-reports, which can be time consuming for participants. Researchers are exploring semi-supervised methods to reduce the need for labeled data and improve emotion recognition research [34].

In a systematic review [35], the authors investigate the relationships between the measured signals and emotions using wearable devices. During the study, the top 20 emotions were identified and analyzed. Electrodermal activity (EDA) emerged as the most frequently adopted signal, measuring 16 emotions. Additionally, the heart rate (HR) was commonly used to assess “positive,” “negative,” “engagement,” “stress,” and “peacefulness” emotions, while skin temperature (ST) and pulse measurements were utilized in some instances. Researchers employed five distinct approaches in studying emotions based on data from wearable devices: machine learning, inferential statistics, deriving emotions from physiological values based on previous literature, descriptive statistics, and specialized algorithms. Inferential statistics, such as correlation and analysis of variance, were used to determine statistically significant relationships between physiological signals and emotion measurements.

Moreover, emotions were derived from physiological values based on conclusions from earlier literature. Descriptive statistics were employed to compare educational emotions measured by wearable devices with the results obtained from more established approaches. Lastly, specialized algorithms were utilized for specific purposes, such as estimating positive and negative emotions based on heart rate spectral analysis. The review overviews the connections between educational emotions and wearable devices in diverse teaching and learning contexts.

### 2.8. Stimuli

Researchers in previous studies consider multiple stimuli to detect mental stress in individuals. According to a study conducted by Guk et al. (2019), different parameters are used to analyze stress levels, and effective videos, music, and sounds are often used to elicit emotional changes in individuals. These stimuli are easy to use and readily available in most databases and research. Videos and audio are selected because they elicit the most response from individuals, and their responses can be detected and monitored spontaneously [36]. Some databases and research articles also use different stimuli, which can be applied in real-life experiences and examples, such as playing physical games, solving math problems, learning, and walking around the city. These methods provide a detailed assessment and analysis of individual responses toward everyday activities and regular tasks being performed [36].

### 2.9. Effects of Emotions on Mental Well-Being

Another set of research studies and literary sources proposed the fact that there is a strong and influential relationship between mental well-being and emotional changes within a person. According to Kumar et al., the emotions which largely influence an individual’s mental state, can be categorized into six major categories: fear, disgust, joy, anger, sadness, and surprise. However, stress has recently been added to these emotions, which can be defined as a person’s mental state, which usually arises when a person has been subjected to unexpected or unbearable circumstances, which are beyond their capacity to tolerate and handle properly [37].

Guy et al. revealed some assessment methods for individuals regarding their mental well-being or presence of stress or anxiety. The most popular and widely used invasive method has been used by healthcare practitioners, where blood cortisol levels are measured and analyzed by physicians, but to modify the detection strategies and methods, scientists and researchers have also proposed some non-invasive methods. These methods include the detection of brainwaves with the help of EEG electrodes or through assessment of biomedical tools to detect physiological biosignals [38].

### 2.10. Use of EEG Electrodes

Certain devices and equipment have been formulated by different researchers, with the help of which wearable devices can conduct an EEG of a person spontaneously and immediately provide the result to healthcare professionals or the individual themselves [39]. Although EEG electrodes provide effective results and measurements regarding an individual’s electrical impulses and bearing activities, the method also possesses a major disadvantage. The electrodes, which are used for the detection of brain activities, need to be attached to the scalp of a person, which can sometimes become painful, disturbing, uncomfortable, or inconvenient for the person; therefore, despite the accuracy and precision, which is being achieved with the help of this method, it cannot be used on a large scale because of its inconvenient monitoring and measuring methods [40]. Instead of EEG electrodes, the most beneficial and advantageous devices—largely preferred for detecting mental well-being and preferred by healthcare providers—are compound semi-conductor (SC) transistors. These are wearable devices, which are responsible for analyzing and monitoring the skin conductivity of a person. However, it has also been reported that most of these devices, which help detect skin conductivity, are not considered for detecting the components and constituents of sweat produced by the human body. The amount of cortisol produced within the sweat of an individual is considered one of the most effective parameters largely used for the detection of stress levels within a person [41]. In addition to the stress hormone cortisol, additional components are also being investigated and examined within the sweat of the human body, which are named volatile organic components (VOCs); different gases are also being released from the human body, and they are sufficient for depicting the current mental health and situation of a person [42].

### 2.11. Devices for Mental Well-Being in Workplaces

Multiple research studies have reported that mental stress is frequently observed in workplaces due to overwhelming challenges and issues faced by individuals. Consequently, appropriate solutions must be provided to employees and staff members to monitor their mental conditions and offer effective remedies as necessary. Heart rate variability, electroencephalographic data, and electrodermal activities are essential for detecting mental stress and anxiety, and they are emphasized in this research study. Li et al. [13] reported that heart rate variability shows the balance and capacity of the autonomic nervous system, which is often used in the workplace to analyze workload and stress development. Biosignals such as heart rate variability are preferred for monitoring stress levels within individuals, as discussed below.

#### 2.11.1. EEG

Electroencephalography (EEG) is a non-invasive method used in neuroimaging, wherein electrical brain activity is measured from the scalp. This technique offers valuable insights into brain functioning by recording the electrical signals neurons generate. It is widely recognized as one of the most used and accessible in vivo neuroimaging methods owing to its ability to provide real-time data and user-friendly nature. EEG finds applications in various domains, including neuroscience, diagnosing neurological disorders, sleep research, and cognitive studies. It has emerged as a potent tool for comprehending brain activity and has significantly contributed to advancing our knowledge of brain functions and dysfunctions [43]. However, multiple challenges and complications have also been reported by the researchers and participants of these studies. Primarily, the major issue being reported is that proper electrode preparation and accurate placement on the scalp are essential for obtaining accurate EEG data. Incorrect placement can lead to poor signal quality and inaccurate results. Participants wearing EEG caps may experience discomfort due to the weight and tightness of the cap, which can affect their natural behavior and potentially influence the signals recorded [44]. Despite these issues and challenges, it has been reported that certain wearable devices have also been introduced, where dry EEG is conducted, and they have been reported to provide effective and beneficial results and outcomes in detecting the mental stress of a person [45,46,47].

#### 2.11.2. Electrodermal Activity

Researchers have introduced multiple wearable devices for detecting electrodermal activity (EDA) in individuals, providing accurate and hindrance-free results. EDA is analyzed to determine the differences in dermal activities and the individual’s constitution, with changes in dermal temperature and cortisol secretion observed in response to stressful conditions. EDA signals are categorized into skin conductance levels (SCL) and skin conductance responses (SCR). SCL is the tonic component, and SCR is the component detected in response to a stimulus. The amount of cortisol in sweat and overall dermal temperature are measured in both conditions, as they are expected to increase during stressful situations [48,49].

#### 2.11.3. Heart Rate Variability 

Assessing heart rate variability has been reported to be a significant component in detecting mental stress levels in individuals. Electrocardiogram (ECG) reports are commonly used to determine the heart rate of an individual, which provides the monitoring measurements of vital signs, such as the blood volume pulse (BVP), the electrical activity of the heart, and the blood flowing in peripheral vessels. Heart rate variations are calculated and determined with the help of BVP readings. HRV is usually calculated based on high- and low-frequency bands, and an increase in the LF and LF/HF ratio has been linked to high anxiety, stress, and anxiousness among individuals. Heart rate variation is usually increased due to overstimulation of the autonomic nervous system or activation of emotional distress, which helps determine the stress levels within an individual over a specific time interval [45,50].

### 2.12. Analysis and Summary 

The literature review presented in this paper explores various strategies for stress identification using wearable physiological sensors. Differences between the approaches include the study design (laboratory experimentation or real-life environment), methodological approach, and use of varying physiological signals. Two classes of techniques are considered regarding the study design: those utilizing laboratory data without investigating the efficiency of the method in a real-life scenario; and those established based on real-life data, but which may have limitations in detecting psychological stresses due to inherent limitations in describing the ground truth. Machine-learning algorithms dominate the applied methods in the literature review. Various combinations of physiological signals were examined, often associated with the spatial qualities of the physical condition and occasionally upheld by video recordings. The literature review suggests that fewer attempts have been made to combine the two approaches for detecting stress levels in a well-controlled laboratory environment and extend that knowledge to a real-life setting [51]. Nonetheless, there is still a need for procedures which can accurately identify stress in individuals in a real-life context. The identified research challenges resulting from the literature survey and study are detailed in the following section.

## 3. Study Design and Framework for Utilizing Wearable Technologies in Enhancing Mental Well-Being

### 3.1. Research Challenges

The literature review presented in the previous sections highlights several strategies for stress identification utilizing wearable physiological sensors. Based on the literature survey, the identified research challenges are detailed in the following section. These challenges aim to assess the use of multi-modal wearable devices in capturing data of various formats to explore assistive solutions for mental well-being; correlate the captured wearable data with gold standard and conventional tests, mainly based on face-to-face assessment and questionnaires; and apply machine-learning methodology for identifying stress factors in heart rate variability, dermal temperature, and EEG signal measurements on data obtained from wearable devices.

### 3.2. Research Objectives 

The major objectives of this research study are.

To analyze and investigate the most renowned wearable tracking device, which is largely prevalent among patients.To assess the effects of these wearable devices on the mental and psychological aspects of patients.To observe literary sources and the recent research and modifications regarding wearable tracking devices.To carry out controlled and open studies applying tested and validated wearable-based measurement settings.To utilize wearable devices for implementing techniques and formulae for measurement.To use wearable devices for achieving methodologies as well as algorithms for measuring mental wellness and feedback.To test the trained model in a real-life scenario, biological signals will be monitored for assembling data on, e.g., heart rate variability, dermal temperature, and EEG signals.

### 3.3. Research Hypothesis

Wearable tracking devices help provide solutions for mental wellness based on the belief that these devices can provide individuals with valuable data and insights, which can help them make positive changes to their lifestyle and improve their mental health. However, further research is needed to fully understand the potential benefits of wearable tracking devices for mental wellness and to identify the most effective ways of utilizing these devices to support mental health.

### 3.4. Research Methodology for Systematic Literature Review

The research study for this project primarily utilized an interpretive systematic literature review design. Additionally, quantitative analysis was adopted to analyze the relevant research and results. For this purpose, a questionnaire survey was distributed among the selected participants to acquire their reviews, perceptions, and knowledge of tracking devices. The survey responses were analyzed and observed using an Excel sheet to illustrate the results in the form of graphs, tables, and charts.

The study collected primary data from electronic databases, including MDPI, ACM Digital Library, Science Direct, and WebMD, APA, PMC, and IEEE, which are reliable and authentic sources for research in the information technology field. Some of the citations from selected research articles were also manually searched. 

This research study aims to emphasize the importance of monitoring and detecting stress and anxiety levels among individuals for their mental well-being and diagnosis by physicians. The systematic literature review methodology utilized for this study includes case–control studies, open studies, and cohort studies of selected participants and samples. Previous research articles that conducted experiments provide detailed insights and information about the progress, modifications, and further innovations in wearable devices used to detect stress among participants. The study is designed systematically to cover all aspects and research in ascending order of year to properly trace and highlight modifications in wearable devices.

### 3.5. Inclusion Criteria 

We only included studies published in English and published between 2010 and 2023 to ensure that the review includes recent advancements in wearable technology for mental well-being. We also included studies focusing on the use of wearable technology for mental well-being, such as stress, anxiety, depression, mood, emotion recognition, and sleep tracking. This criterion is important because the review aims to evaluate the effectiveness of wearable technology in providing assistive solutions for mental well-being. Studies that utilized quantitative, qualitative, or mixed-method research designs are also included in the review.

### 3.6. Exclusion Criteria

Studies not published in the English language or published before the year 2010 were excluded from the review. Additionally, studies that did not focus on wearable technology for mental well-being or did not evaluate the effectiveness of wearable technology for mental well-being were excluded. Finally, animal models or theoretical studies were excluded from the review.

### 3.7. Research Onion

Saunders et al. [52] proposed the research onion as a method in which different layers and stages of research are utilized and adopted. The research onion process provides an improved method for designing, planning, and executing each layer of research in a stepwise manner. By utilizing the research onion approach, better outcomes and results can be achieved in a properly designed manner [53].

### 3.8. Research Philosophy

The research philosophy adopted for writing this structured literature analysis is the idea of epistemology [54]. This approach was chosen for certain types of research, which require proven experimental analysis to be conducted. The procedure aims to validate the research priorities and theories, since research that was already carried out is primarily used in the analysis.

### 3.9. Research Design

A.Positivism was chosen as the research design for this study, but certain strong characteristics of constructivism and objectivism may also be employed because the selection is dependent on the research priorities, concerns, and theories. However, considering the specific orientation of this study, positivism was chosen as the preferred research design.

### 3.10. Research Approach

The third layer of the research onion primarily helps classify the method employed during the compilation of multiple datasets. The deductive method is preferred for this research analysis, since the hypothesis was formulated and implemented to conduct the systematic literature review. All libraries and academic papers were obtained based on this theory.

### 3.11. Research Strategies

The survey-based methodology using the deductive approach was employed as the testing method for this review. In addition to the thorough examination and review of databases with specific keywords, a survey analysis was also conducted for the research report to gather people’s awareness, details, and insights regarding the idea of tracking devices [8].

### 3.12. Research Choice

B.A multi-method research design was selected for this study, as the related studies and review papers are the databases, which are used and analyzed in this research analysis. Accordingly, quantitative research was conducted while analyzing and classifying all related literary sources [55]. The literary sources were classified, skimmed, and selected using formulae and Excel sheets based on quantitative analysis.

### 3.13. Time Horizon for the Research Study

A linear time frame was chosen for this analysis, as various libraries, literary materials, and academic papers will be researched and evaluated repeatedly over an extended period [50,56].

### 3.14. Research Analysis and Data Collection

C.The final layer of the research onion [56] primarily displays the various phases of scientific study and data collection techniques. To assess its reliability, validity, and accuracy, the data collected and accumulated so far were further analyzed.

### 3.15. Sample Size

Almost 100 papers were selected for the research review based on the previously mentioned exclusion and inclusion criteria. However, in our research report, not all research papers were emphasized, and only the most important papers were finalized after extensive skimming and thorough reading of each post. Sixty-one papers were chosen during the skimming process, providing a full understanding of the current literature on wearable tracking devices primarily used for human physical and mental observation.

### 3.16. Sample Ethics

A research survey was administered along with the literary tools for the research project. The participants were recruited through an email with a link to a Google Form to answer some questions. The team aimed for 21 participants—a number proven sufficient for providing information for the proof of concept in previous studies. The participants were healthy postgraduate students at the Queen Mary University of London aged 18 to 45, comprising 47.6% males and 52.4% females. An ethical consent form was given to these participants, emphasizing that all the knowledge and answers gathered from them would be used exclusively for research.

### 3.17. Data Analysis

The data for the study were collected using the Google Forms service, then coded and processed with Microsoft Excel, as well as the Statistical Package for the Social Sciences (SPSS) version 23. The internal validation of the scales was approved during the study.

Demographic Factors

As shown in Table 1, 21 people participated in the current study, including 11 (52.4%) females and 10 (47.6%) males. Their age was between 18 and 45, with an advantage for the 25–34 group (81%). They all declared that they are healthy and have no chronic disease.

2.Wearable Device Status

As shown in Table 2, only 38.1% of participants used wearable devices, and over 60% said they had never used them to communicate with friends, family, or doctors.

3.Reliability of the Study Scale

The Pearson correlations were positive and significant, ranging from (*r* = 0.87, *p* < 0.01) to (*r* = 0.24, *p* < 0.05), and Cronbach’s alpha coefficient achieved great value (α = 0.77). Therefore, the scale was valid and reliable (Table 3). 

4.The Perception Toward Wearable Devices and the Relationship with Gender and Users

Table 3 displays the mean score for the perception of using a wearable device (3.67 ± 1.07). There was an insignificant difference in the mean score in terms of gender and wearable device users and non-users (*p* > 0.05).

5.Purpose of Using Wearables

As shown in Table 4, 66.67% of participants indicated they used a wearable device to monitor physical activities, and 52.63% used it to monitor daily activities.

Regardless of the variety of stressors, which may occur, the physical reactions to stress are similar [25]. For biological, physiological, sociological, and philosophical stressors, the body’s stress response is the same. Stress causes the body to increase the heart rate, blood pressure, and muscle tension, as well as to increase glucose and serum cholesterol production while decreasing protein stores, digestive processes, and T-lymphocytes. There is a lot of evidence that stressful life events and perceived stress are linked to immune system changes. Chronic physiological responses to stress can lead to illness or disability. For example, HRV feedback reflects the activity of the autonomic nervous system’s sympathetic and parasympathetic branches. This enables the development of strategies to gain voluntary control of emotional regulation. For sports performance, biofeedback training has demonstrated some psychological effects, such as the reduction in anxiety and self-control enhancement. It is also a tool for improving the prevention and treatment of overtraining and athletic injuries. The above results show that wearable devices are assistive in providing health and mental wellness solutions. Because physical activity tracing would reduce the stress stressors affecting the subject’s body, therefore, it would also affect mental activity and reduce cognitive mental load.

### 3.18. Limitations

Therefore, due to the extremely small number of scientific papers and experiments on the subject carried out so far, there is not a large amount of information available on the various forms of wearable monitoring systems, which may be used for the mental analysis of a human being. It is possible to calculate a person’s physical vital signs and parameters—and their different variations—using wearable devices, but they are not very significant in measuring human cognitive behaviors.

### 3.19. Preliminary Study Results and Analysis

Selection and Preparation of Database

Multiple research works and theories were identified and included in this research study; however, to form a systematic literature review, a certain pathway/framework was followed for this purpose, which is shown in Figure 1.

The systematic literature review was conducted in a specific order, as shown in the chart. Initially, 100 research articles were considered for the study based on the selected keywords, such as wearable devices, mental health assessment, mental health in the workplace, and advanced wearable technologies. The articles were then screened based on their year of publication, and only those published after 2010 were included in order to rely on recent research regarding the research topic. However, the subject’s origin was also important to consider, and some theories, histories, and backgrounds from previous research were included but not relied upon completely. It should be noted that wearable devices have been used to assess physical and biological functions for many years, but their use for mental health assessment has only been introduced recently. 

To further investigate the research topic, several research articles were identified based on the selected keywords and year of publication. In addition to this, the authors of research articles were also selected based on their affiliation with the academic institutes and research centers. The most recent research articles were focused on obtaining a more precise and accurate result. The literature review and research articles included in this paper revealed that, despite the abundance of information, there are still gaps in the research regarding the use and implementation of wearable devices in the workplace. These areas require further attention and investigation in future studies.

2.Results and Findings

Research studies reported that wearable tracking devices have become essential for constantly measuring and monitoring the performance, physical activities, behavioral changes, and changes in the mental health patterns of an individual. This is particularly important in monitoring the effects of different stimuli on an individual’s mental health and well-being. In addition to physical well-being, workplaces and organizations have begun to focus on individuals’ mental well-being and health to ensure their productivity and creativity levels remain stable [57]. To further intensify research studies, authors and researchers have considered certain parameters and factors to be investigated and emphasized further in future research. 

The most common parameters used to analyze mental health are the presence of cortisol in a person’s sweat, heart rate, blood pressure, body temperature, and pulse rate. Evaluating these parameters helps identify anxiety and stress among individuals within workplaces. To analyze the differences in their biological and mental behaviors toward such stimuli, individuals are also provided with certain stimuli, such as stressful conditions. Some of these factors include the provision of excessive workload for employees, challenging situations involving mind games, limited time for performing complex activities, and conflicting situations. With the help of research, it was also identified that there are additional terms widely used to assess which type of monitoring, recording, assessment, evaluation, gathering, and measurement of the information of an individual is being performed in terms of their performances, physical or biological behaviors, responses, and environmental data, and the responses and readings are acknowledged for further research.

## 4. Discussion

The aim of the manuscript is to coordinate and conduct a comprehensive systematic literature review, which includes a thorough evaluation and study of wearable technologies for detecting mental well-being and human cognitive effectiveness. While some studies highlighted the use of these technologies in the workplace, there is still a range of challenges in applying these strategies, which need to be investigated by the researchers. In some papers included in the research review, it was stated that the total number of these articles was split into two key categories. Numerous academic papers focused on using and evaluating wearable technologies in laboratory environments; no real-life results were included in these research studies. On the contrary, it was also mentioned that a significant number of research papers examined real-life evidence rather than laboratory experimentation; therefore, it is possible to suggest the systems and devices, which could be conveniently used within the workplace [29]. Analysis methodologies, which adapt data from structured environments within the laboratory, are generally inadequate for delivering reliable readings, including equipment findings, which can be reliably and accurately regarded. On the contrary, it was also stated that methodologies where real-life data were considered produced findings with significant specificity and consistency, as they were capable of measuring stress and anxiety within an individual under various environmental conditions. Based on the study reports, it was also found that it is important to concentrate on the operating process and processes of the technologies being used to maintain the results of the findings correct and consistent. Several of the sensor-based portable devices were also reported in a variety of studies to be highly inadequate and unable to provide reliable data and readings on body temperature, variability of heart rate, and human skin performance. These three criteria are important for evaluating a person’s stress levels and anxiety disorders, and they largely help provide reliable results [13]. Of these three, the most genuine and important element is the presence of cortisol in the individual’s sweat. The more increased the level of cortisol, the greater the level of physiological arousal. Humans employ multiple devices and equipment to primarily measure these three vital signs. Empatica E4, Microsoft Band 2, Samsung Gear S, Body Media Sense Wear Armband, and Neurosky MindWave are the wireless products, which are often used and considered for this purpose. This occurs especially when the sensors assembled and used in wearable devices are not effective and not of better quality. Inaccurate findings are collected because of errors in the sensors of these wearables, which also leads to errors in the research results. The investigator does not adequately control stress and anxiety levels due to problems with this equipment and the readings collected [58]. To implement an adequate methodology for comprehensive research of wearable devices for diagnosing mental well-being and cognitive awareness, researchers need to concentrate primarily on the four key areas, including a successful study design, recruitment of participants, choice of wearable devices, and techniques for data analysis.

Stress is considered one of the major components of mental health, which results in multiple other physical and psychological problems. It has also been reported in numerous research studies that stress management must be considered an important and essential element within workplaces and organizations, so that employees can work with increased enthusiasm and motivation [59]. Figure 1 mainly shows some of the important parameters and vital signs, which have to be observed and deeply investigated to analyze the different types of changes within a person. For example, the help of eyelid movements, changes in facial expressions, spontaneous and abrupt head movements, and pupil dilation are some of the important physical signs, which indicate stress or anxiety within an individual. In difficult or complex situations, a person usually performs rapid head movements, or the pupils are dilated. In addition, an abrupt eyelid movement is observed, and most importantly, the individual’s facial expressions are rapidly changed [27].

However, the objective of our research study was to analyze the effectiveness of wearable devices for detecting stress levels. Therefore, it was observed that, in most cases, the parameters which are often monitored with the help of devices are GSR, BT, and PPG, and these vital signs are monitored using ECG, EEG, and skin conductance measurement devices. In addition, heart rate variability is also monitored with the help of these devices; however, body temperature can be monitored simply with the help of a thermometer [23].

Another aspect being proposed in the reports is that to improve the mental well-being of an individual, it is also important to indulge them in daily physical activity, as this largely helps relieve stress to a considerable extent. Several wearable devices are used for this purpose as well, which help in the detection of body weight, obesity level, calorie intake, dietary habits, and other similar parameters of an individual [24]. With the help of these wearable devices, it becomes convenient and feasible for individuals to indulge in some sort of physical activity to promote their mental health and well-being. Patients presenting with serious mental illness tend to lean toward weight gain and obesity twice as much as normal individuals [59,60]. Therefore, to keep the body physically fit and healthy, it is essential to keep the psychological conditions of the body intact and stable as well [25].

A statistical research report concluded that more than 300 million people are diagnosed with low or high degrees of stress and anxiety at some stage of their life. The increasing number of people with anxiety and stress is also leading to severe downturns for business organizations and the countries’ economies. Depression and anxiety are some of the most common psychological disorders, which have become largely prevalent in the global world [26]. The overall cost being incurred as a result of expenditure on the betterment and treatment of patients with stress and anxiety disorders stands at more than USD 2.5 trillion [60,61]. Most importantly of all, one of the major problems revealed is the shortage of doctors, physicians, psychiatrists, and healthcare providers, which results in the lack of monitoring of different vital signs and parameters using microelectromechanical, biological, and chemical sensing, as well as electrocardiogram (ECG)-, electromyogram (EMG)-, and electroencephalogram (EEG)-based neural sensing. One of the essential advantages of these wearable devices is that they are considerably easier to use and cheaper than the difficult and complex instruments used in the laboratories of science and health institutions for monitoring the biomarkers of the human body [29].

Several instruments and devices introduced by various companies include ACCU-CHECK by Roche Diagnostics, iSTAT by Abbott, and Lactate Scout by Sport Resource Group. The headquarters of these companies are located as follows: Roche Diagnostics (ACCU-CHECK) in Basel, Switzerland; Abbott (iSTAT) in Abbott Park, Illinois, United States; and Sport Resource Group (Lactate Scout) in Minneapolis, Minnesota, United States. These devices are usually launched and presented on the consumer market for commercial purposes. They are mainly intended to measure the vital signs of athletes and sports individuals, so that their performance and activities can be improved. However, these vital signs can also be used for stress level detection and analysis of anxiety within individuals. It has also been reported that these devices usually take blood samples of an individual for detection of all these vital signs, which is a time-consuming process and considered a huge barrier and challenge for sports companies [60].

## 5. Conclusions

The paper highlights the positive impact of IoT on various fields, including healthcare, which has been positively affected due to the development of wearable devices, which detect various parameters, biomarkers, and biosensors to evaluate stress. Multiple types of wearable devices are largely used, such as ECG, EEG, EMG, PPG, and BT, which help in detecting the galvanic skin conductance, heart rate variability, and body and skin temperature of an individual. The paper also highlights the challenges and issues faced by participants and researchers in the implementation of wearable devices. These challenges include the difficulty for participants in wearing EEG patches and understanding the overall mechanism and working of these devices. Despite the challenges, wearable devices have facilitated the detection of stress and anxiety levels, which helps in providing appropriate strategies and a safe working environment to enhance the productivity and creativity of employees. Overall, the report suggests that wearable devices have the potential to play a crucial role in maintaining mental well-being in the workplace.

## Figures and Tables

**Figure 1 sensors-23-07378-f001:**
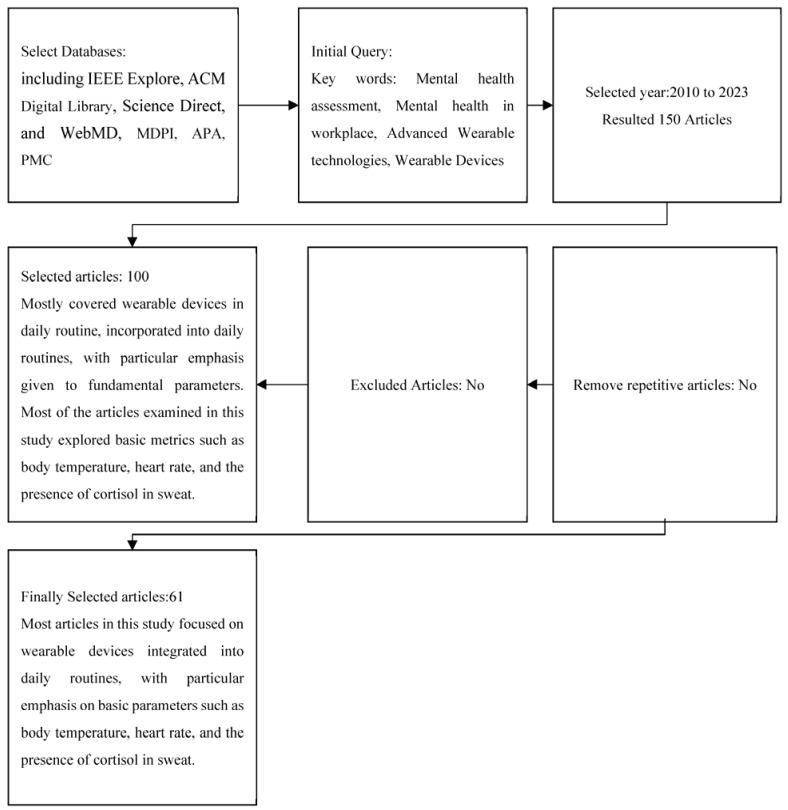
Systematic literature review framework.

**Table 1 sensors-23-07378-t001:** Demographic characteristics of the participants (N = 21).

Demographic Factors		Frequency	Percent
Gender	Male	10	47.6
	Female	11	52.4
Age	18–24	2	9.5
	25–34	17	81
	35–44	2	9.5
Health Status	Excellent	9	42.9
	Very Good	11	52.4
	Fine	1	4.8

**Table 2 sensors-23-07378-t002:** Status of wearable devices.

Wearable Device Status		Frequency	Percent
Do you use wearables or not?	Yes	8	38.1
	No	13	61.9
Using wearable devices to communicate with: Friends	Daily	2	9.5
	Several Times a Week	5	23.8
	Never	14	66.7
Using wearable devices to communicate with: Family	Daily	5	23.8
	Several Times a Week	2	9.5
	Seldom	1	4.8
	Never	13	61.9
Using wearable devices to communicate with: Doctors	Daily	1	4.8
	Once a Month	2	9.5
	Seldom	4	19
	Never	14	66.7

**Table 3 sensors-23-07378-t003:** Perception toward wearable devices and the relationship with gender and users.

Statement	Mean	Std. Deviation	r	Gender t/*p* Value	Users t/*p* Value
1. Wearable devices motivate me to evaluate my state of mind.	3.17	0.83	0.40 *		
2. The feedback given via wearable devices is useful.	2.92	1.24	0.69 *		
3. Sharing my data measured with wearable devices encourages me to evaluate my mental wellness.	2.42	1.24	0.65 *		
4. I feel happy with a software, which can assess me using wearable devices, which will measure my biomarkers (such as HRV, EEG, skin conductance).	2.83	1.03	0.24 *		
5. I feel that tips evaluating the level I reached in my emotional assessment would be conducive to my mental wellness.	4.00	1.13	0.87 *		
Overall	3.67	1.07		t = −1.70/0.12	t = −1.09/0.30
Cronbach’s Alpha		0.77			

* means significant at (0.05).

**Table 4 sensors-23-07378-t004:** Purpose of using wearables.

Statement		Frequency	Percent
**You use wearables to**	Communicate with people	1	11.11
	Monitor physical activities	6	66.67
	Monitor mental wellness	2	22.22
**What kind of feedback would you like to get from your wearables?**	Monitor daily activities	10	52.63
	Advice on how to tackle daily challenges	1	5.26
	Feedback on general wellness	2	10.53
	Improving your wellness in general	5	26.32
	None of the above	1	5.26

## Data Availability

Not applicable since all the data either stated in the manuscript or available from the literature as this is a systematic review study.

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
