# Peer review of "The Use of Wearable Technology in Providing Assistive Solutions for Mental Well-Being"

_sensors, 2023, doi:10.3390/s23177378_

Round 1

Reviewer 1 Report

Dear authors,

Introduction:

Line 96. ‘Due to disorders’.

Lines 210 to 229 are the same as the following 2 paragraphs.

2.4. ASSESSMENT OF EMOTIONS

In this chapter, there should be more information about the impact of emotions. There is a recent review where they compile many studies talking about how emotions can affect cognition. I suggest  finding another reference to describe more.

2.11.1. EEG

“The objective of using an Electroencephalogram is to detect the blood flow within the head region of an individual, as it is considered one of the most promising bio-signals for the detection of stress levels within an individual. Research has revealed the fact that due to increased levels of stressful conditions, the blood flow within the brain of a person is increased, which also results in increased body temperature of the individual.” – I’m afraid that this is fNIRS and not EEG. Electroencephalography, as the name suggests, measures electrical activity. Correct this paragraph.

“In addition, it has also been reported that it is not easy to constantly equip such devices on the head of a person in their workplaces, and at the same time, they are expensive and unaffordable for most laboratory experimentation procedures, as well” – One more time, I disagree. You cited a study, but are they actually saying that? Because EEG is known to be one of the cheapest equipment for evaluating the brain and it’s portable. If they find EEG expensive, I hope they never need to do a fMRI.

3. STUDY DESIGN AND STRUCTURE FOR WEARABLE TECHNOLOGIES IN MENTAL WELLBEING

The title needs to be corrected in wearable.

In line 56, you say that you examined 50 from 2016 to 2021, then in line 490, you mention that the inclusion was between 2010 and 2022, and finally, in line 550, you said that only 40 papers survived. Moreover, in Figure 1 (in my opinion, this figure should be at the beginning), we have an outcome of 58 studies… Be consistent, please.

From lines 605 to 607, there needs to be a reference. I suggest **Domingos et al., 2021. These authors showed that biofeedback (neurofeedback) training improves HRV.  

** Domingos, C., Silva, C. M. D., Antunes, A., Prazeres, P., Esteves, I., & Rosa, A. C. (2021). The influence of an alpha band neurofeedback training in heart rate variability in athletes. International Journal of Environmental Research and Public Health, 18(23), 12579.

4. Discussion

The title shows 3. Discussion. Correct it, please.

In the first line, delete “key”.

Line 688. A period is missing.

So, several devices and equipment are used by humans to test these mainly three vitals, so Empatica E4, Microsoft Band 2, Samsung Gear S, Body Media Sense Wear Armband, Neurosky Brain Wave, and XYZLL are the wireless products that are often used and considered for this purpose.” – Please, avoid the wording “so” in a research paper.

5. Conclusions

The title shows 4. Conclusions. Correct it, please.

Good luck to you.

Already on the commentaries for the reviewers. 

Author Response

Original Manuscript ID: sensors-2536913
Original Article Title: “The Use of Wearable Technology in Providing Assistive Solutions for Mental Wellbeing”
To: MDPI Sensors
Re: Response to reviewers
Dear Editor,
We would like to extend our warmest thanks and appreciation to you, the editorial team and the reviewers for allowing us to resubmit our manuscript addressing the comments and concerns raised by the reviewers.
We are uploading (a) our point-by-point response to the comments (below) (response to reviewers), (b) an updated manuscript with yellow highlighting indicating changes (under “Author’s Response Files”), and (c) a clean updated manuscript without highlights (“Main Manuscript”).
Best regards,
Reham Alhejaili
on behalf of the authors
Reviewer#1,
We would like to extend our appreciation and thanks to the reviewer for their valuable feedback and suggestions that helped us in improving the quality of the manuscript. We have addressed the raised comments point by point in the following section. Introduction: Line 96. ‘Due to disorders’. Author action: We have considered the suggestion and updated the manuscript accordingly. Lines 210 to 229 are the same as the following 2 paragraphs. Author action: Apologies for this error and we have now removed the repetition form the revised manuscript. 2.4. ASSESSMENT OF EMOTIONS In this chapter, there should be more information about the impact of emotions. There is a recent review where they compile many studies talking about how emotions can affect cognition. I suggest finding another reference to describe more. Author action: We agree with the reviewer, and we have added a reference in the updated manuscript addressing this exact comment. 2.11.1. EEG “The objective of using an Electroencephalogram is to detect the blood flow within the head region of an individual, as it is considered one of the most promising bio-signals for the detection of stress levels within an individual. Research has revealed the fact that due to increased levels of stressful conditions, the blood flow within the brain of a person is increased, which also results in increased body temperature of the individual.” – I’m afraid that this is fNIRS and not EEG. Electroencephalography, as the name suggests, measures electrical activity. Correct this paragraph. “In addition, it has also been reported that it is not easy to constantly equip such devices on the head of a person in their workplaces, and at the same time, they are expensive and unaffordable for most laboratory experimentation procedures, as well” – One more time, I disagree. You cited a study, but are they actually saying that? Because EEG is known to be one of the cheapest equipment for evaluating the brain and it’s portable. If they find EEG expensive, I hope they never need to do a fMRI.
Author action: Out intention was to describe the cause of the captured data and signal changes and link those to change in the actual biophysical parameter variations, but we do appreciate that this did not come cross clearly and therefore we have updated this section accordingly. When we stated that it is expensive and less convenient, we meant to refer to the fact that in comparison to the cost-effective alternatives available now in terms of wearable devices and consumer ones. However, we do understand that the paragraph did not convey our message appropriately and therefore revised accordingly. 3. STUDY DESIGN AND STRUCTURE FOR WEARABLE TECHNOLOGIES IN MENTAL WELLBEING
The title needs to be corrected in wearable.
Author action: Action taken. In line 56, you say that you examined 50 from 2016 to 2021, then in line 490, you mention that the inclusion was between 2010 and 2022, and finally, in line 550, you said that only 40 papers survived. Moreover, in Figure 1 (in my opinion, this figure should be at the beginning), we have an outcome of 58 studies… Be consistent, please.
Author action: This section incorporates and analyzes the findings from previously cited research papers to further enhance our understanding of the topic at hand. By examining the works of other scholars, we aim to gain valuable insights and strengthen the validity of our study. We have updated the manuscript and ensured consistency as suggested. From lines 605 to 607, there needs to be a reference. I suggest **Domingos et al., 2021. These authors showed that biofeedback (neurofeedback) training improves HRV. ** Domingos, C., Silva, C. M. D., Antunes, A., Prazeres, P., Esteves, I., & Rosa, A. C. (2021). The influence of an alpha band neurofeedback training in heart rate variability in athletes. International Journal of Environmental Research and Public Health, 18(23), 12579.
Author action: We have updated the manuscript and modified the references as kindly suggested by the reviewer. 4. Discussion The title shows 3. Discussion. Correct it, please.
Author action: It has been corrected. In the first line, delete “key”.
Author action: action taken. Line 688. A period is missing.
Author action: Period added. “So, several devices and equipment are used by humans to test these mainly three vitals, so Empatica E4, Microsoft Band 2, Samsung Gear S, Body Media Sense Wear Armband, Neurosky Brain Wave, and XYZLL are the wireless products that are often used and considered for this purpose.” – Please, avoid the wording “so” in a research paper. Author action: We have considered the suggestion and updated the manuscript accordingly. 5. Conclusions
The title shows 4. Conclusions. Correct it, please. Author action: Corrected.

Reviewer 2 Report

The article gave a systemic review on the use of wearable technology in providing assistive solutions for mental wellbeing. Overall, it is well drafted. It can be improved by the following aspects:

1. The related references should be added after 2022.

2. In the Abstract, Line 18-19, the authors wrote that “It is also a tool for improving the prevention and treatment of overtraining for athletic injuries”, but in the article, the use of wearable devices in athletic injuries were not involved and included.

3. Various wearable devices were used to monitor the physiological and psychological signals of the wearers. In Line 278-283 the authors introduced many types of the wearable devices. What are the differences of these devices? What signals can these devices detect? These information should be included.

4. 21 participants were recruited for the survey. In Table 2, in can be seen that only 8 participants used wearable devices, so these eight subjects seemed a little small sample for the survey.  

Author Response

Original Manuscript ID: sensors-2536913
Original Article Title: “The Use of Wearable Technology in Providing Assistive Solutions for Mental Wellbeing”
To: MDPI Sensors
Re: Response to reviewers
Dear Editor,
We would like to extend our warmest thanks and appreciation to you, the editorial team and the reviewers for allowing us to resubmit our manuscript addressing the comments and concerns raised by the reviewers.
We are uploading (a) our point-by-point response to the comments (below) (response to reviewers), (b) an updated manuscript with yellow highlighting indicating changes (under “Author’s Response Files”), and (c) a clean updated manuscript without highlights (“Main Manuscript”).
Best regards,
Reham Alhejaili
on behalf of the authors
Reviewer#2,
We would like to extend our appreciation and thanks to the reviewer for their valuable feedback and suggestions that helped us improve the manuscript's quality. We have addressed the raised comments point by point in the following section.

  1. The related references should be added after 2022.
    Author action: We have updated the manuscript and modified the references as kindly suggested by the reviewer and added further recent ones. 2. In the Abstract, Line 18-19, the authors wrote that “It is also a tool for improving the prevention and treatment of overtraining for athletic injuries”, but in the article, the use of wearable devices in athletic injuries were not involved and included Author action: We have considered the suggestion and deleted this reference to injuries and associated applications. 3. . Various wearable devices were used to monitor the physiological and psychological signals of the wearers. In Line 278-283 the authors introduced many types of the wearable devices. What are the differences of these devices? What signals can these devices detect? This information should be included. Author action: We have considered the suggestion and updated the manuscript accordingly. 4. 21 participants were recruited for the survey. In Table 2, in can be seen that only 8 participants used wearable devices, so these eight subjects seemed a little small sample for the survey. Author action: Thank you for this comment. The survey-based methodology, utilizing the deductive approach, is employed in conjunction with the systematic review, which was the dominant factor in this study. The group of surveyed participants is applied to demonstrate some indications based on users’ feedback and to back up some of the claims made through the systematic review. We agree this number might be considered small in terms of those using wearable; however, we were aiming to represent equivalent percentages to actual wearable users compared to the general population and wanted to give some initial indications at this stage to be taken further for more extensive studies as next steps in our research.

Round 2

Reviewer 2 Report

The authors have made a systematic and extensive review on the use of wearable technology in providing assistive solutions for mental wellbeing. The article is well organized and written and gave a great many information. The authors have made revision according to the reviewers' comments. Therefore, I don’t have any further comments.

There are some editing errors.

The page number at the header of the page is not continuous, from the page of Table 1.

Page 6 Line 607, “introduced recently[]”, the number of the reference number is missed.

Page 14, Line 551, “Wearable users”, “Wearable” should be “wearable”

Page 14, Line 554, “Monitor physical activities”, “Monitor” should be “monitor”.

Author Response

We would like to extend our appreciation and thanks to the reviewer for their valuable feedback and suggestions that helped us in improving the quality of the manuscript. We have addressed the raised comments point by point in the following section.

:
There are some editing errors.
The page number at the header of the page is not continuous, from the page of Table 1. Author action: We have considered the suggestion and updated the pages numbers.
